# Early neonatal mortality and determinants in sub-Saharan Africa: Findings from recent demographic and health survey data

**Tadesse Tarik Tamir**[1]*, **Yirgalem Mohammed**[2], **Alemneh Tadesse Kassie**[3], **Alebachew Ferede Zegeye**[4]

1 Department of Pediatric and Child Health Nursing, School of Nursing, College of Medicine and Health Sciences, University of Gondar, Gondar, Ethiopia, 2 Department of Health system and Policy, College of Medicine and Health Science, School of Public Health, Wollo University, Dessie, Ethiopia, 3 Department of Clinical Midwifery, School of Midwifery, College of Medicine and Health Sciences, University of Gondar, Gondar, Ethiopia, 4 Department of medical Nursing, School of Nursing, College of Medicine and Health Sciences, University of Gondar, Gondar, Ethiopia

* tadestar140@gmail.com

**Data Availability Statement:** The data underlying the results presented in this study are third-party data sourced from the Demographic and Health Survey (DHS) program. Access to the data is

## Abstract

### Introduction

Neonatal mortality during the first week of life is a global issue that is responsible for a large portion of deaths among children under the age of five. There are, however, very few reports about the issue in sub-Saharan Africa. For the sake of developing appropriate policies and initiatives that could aid in addressing the issue, it is important to study the prevalence of mortality during the early neonatal period and associated factors. Thus, the aim of this study was to ascertain the prevalence of and pinpoint the contributing factors to early neonatal mortality in sub-Saharan Africa.

### Method

Data from recent demographic and health surveys in sub-Saharan African countries was used for this study. The study included 262,763 live births in total. The determinants of early newborn mortality were identified using a multilevel mixed-effects logistic regression model. To determine the strength and significance of the association between outcome and explanatory variables, the adjusted odds ratio (AOR) at a 95% confidence interval (CI) was computed. Independent variables were deemed statistically significant when the p-value was less than the significance level (0.05).

### Result

Early neonatal mortality in sub-Saharan Africa was 22.94 deaths per 1,000 live births. It was found to be significantly associated with maternal age over 35 years (AOR = 1.77, 95% CI: 1.34–2.33), low birth weight (AOR = 3.27, 95% CI: 2.16, 4.94), less than four ANC visits (AOR = 1.12, 95% CI: 1.01, 1.33), delivery with caesarean section (AOR = 1.81, 95% CI: 1.30–2.5), not having any complications during pregnancy (AOR = 0.76, 95% CI: 0.61, 94), and community poverty (AOR = 1.32, 95% CI: 1.05–1.65).

available by registering on the DHS website (https://www.dhsprogram.com) and submitting a request. The authors did not have any special access privileges, and others can obtain the same data through the standard process.

**Funding:** The authors received no specific funding for this work.

**Competing interests:** The authors have declared that no competing interests exist.

**Abbreviations:** ANC, Antenatal care; AOR, Adjusted odds ratio; CI, Confidence Interval; CS, Caesarean Section; DHS, Demographic and Health Survey; ICC, Intracluster Correlation Coefficient; MOR, Median Odds Ratio; LLR, Log likelihood Ratio; PCV, Proportional Change in Variance; SDG, Sustainable Development Goal.

## Conclusion

This study found that about twenty-three neonates out of one thousand live births died within the first week of life in sub-Saharan Africa. The age of mothers, birth weight, antenatal care service utilization, mode of delivery, multiple pregnancy, complications during pregnancy, and community poverty should be considered while designing policies and strategies targeting early neonatal mortality in sub-Saharan Africa.

## Introduction

The term "early neonatal mortality" (ENM) refers to the death of a newborn that occurs within the first week of life [1]. Although neonatal mortality has decreased significantly over the past few decades, it still poses a serious problem for the majority of low-income countries [2]. The early neonatal period is responsible for around 33% of all under-five deaths globally, whereas the remaining 67% happen after one week of life over their five-year period [3]. Of the 2.8 million babies that die throughout the neonatal period globally each year, 73% do so in the first week after delivery (early neonatal period) [1, 4].

Globally, 2.7 million babies lose their lives within the first month of life due to infections, preterm birth problems, and birth asphyxia. Ninety-nine percent of newborn fatalities and stillbirths take place in low- and middle-income countries like those in sub-Saharan Africa [5]. Sub-Saharan Africa had the highest newborn mortality rate in 2019 with 27 deaths per 1,000 live births, followed by Central and Southern Asia with 24 deaths per 1,000 live births [6]. A child born in sub-Saharan Africa or southern Asia has a tenfold higher probability of dying in the first month of life than a child born in a high-income country [6].

Only one sub-Saharan African country, Madagascar, was found to have met Millennium Development Goal (MDG) 4 in the year 2015 [7]. The Sustainable Development Goals (SDGs) were established during the United Nations meeting in 2015 to succeed the MDGs. Sustainable Development Goal (SDG) three has set a target to reduce mortality in children younger than 5 years to 25 or fewer per 1000 live births and the neonatal mortality rate to 12 or less per 1000 live births by 2030 [8, 9]. Despite the fact that neonatal mortality has decreased significantly since 1990, further efforts are required to accelerate this progress and meet the SDG objective by 2030 [10].

The lack of basic antenatal care [3, 11], complications during pregnancy [12], birth weight [13, 14], cultural practices [15], wealth index [16], home delivery [17], difficulty affording health care [18], and low maternal education status [19] were among the factors contributing to neonatal mortality in low and middle income countries.

The early neonatal period, the first seven days of life, are the most precarious for a baby's survival [6]. Children face the highest risk of death in their first week of life [6], especially in low- and middle-income countries. To the knowledge of researchers, no study has been conducted on neonatal mortality in the first week of life and determinants at inter-country levels in sub-Saharan Africa. Coming up with its magnitude and determinants helps understand the current status of the problem and can be an input for policymakers. Hence, this study aimed to assess the prevalence and determinants of early neonatal mortality in sub-Saharan Africa.

## Methods

### Data and study setting

The secondary data analysis of recent sub-Saharan Africa (SSA) Demographic Health Survey datasets from 2014 to 2019/20 was conducted (Table 1). The datasets were appended together to investigate early neonatal mortality and determinants in SSA. The data were accessed from the official Demographic Health Survey program database (http://www.dhsprogram.com). The Demographic and Health Surveys (DHS) are surveys that are nationally representative and offer data that is comparable across nations to monitor and assess impact indicators in the areas of population, health, and nutrition. The DHS uses a two-stage stratified cluster sampling technique. The first stage is to select enumeration areas, and the second stage is to draw a sample of households in each enumeration area. We used DHS surveys in 23 sub-Saharan African countries. This study used the children's record dataset to determine the outcome variable. A sample of 262,763 live births was included in the study (Table 1). While all babies born alive from women of reproductive age five years before the surveys in sub-Saharan Africa were the source population, all live births in the enumeration areas of the survey were the study population.

### Measures

The outcome variable for this study was early neonatal mortality, which was dichotomized as "yes" = 1 for neonates who died within one week of life and "no" = 0 for neonates who were

**Table 1. Sample size for early neonatal mortality in sub-Saharan Africa countries (n = 262,763).**

| Country | Year of survey | Frequency | Percent |
|---|---|---|---|
| Angola | 2015 | 14,322 | 5.45 |
| Benin | 2017/18 | 13,589 | 5.17 |
| Burundi | 2016/17 | 13,192 | 5.02 |
| Cameron | 2018 | 9,733 | 3.7 |
| Chad | 2014/15 | 18,623 | 7.09 |
| Ethiopia | 2016 | 10,641 | 4.05 |
| Gambia | 2019 | 8,362 | 3.18 |
| Ghana | 2015 | 5,884 | 2.24 |
| Guinea | 2018 | 7,951 | 3.03 |
| Kenya | 2014 | 20,964 | 7.98 |
| Lesotho | 2014 | 3,138 | 1.19 |
| Liberia | 2019/20 | 5,704 | 2.17 |
| Malawi | 2015 | 17,286 | 6.58 |
| Mali | 2018 | 9,940 | 3.78 |
| Nigeria | 2018 | 33,924 | 12.91 |
| Rwanda | 2019/20 | 8,092 | 3.08 |
| Senegal | 2019 | 6,125 | 2.33 |
| Serra Leone | 2019 | 9,899 | 3.77 |
| South Africa | 2016 | 3,548 | 1.35 |
| Tanzania | 2015 | 10,233 | 3.89 |
| Uganda | 2016 | 15,522 | 5.91 |
| Zambia | 2018 | 9,959 | 3.79 |
| Zimbabwe | 2015 | 6,132 | 2.33 |
| Total sample size | 2014-2019/20 | 262,763 | 100 |

alive within one week of life. Considering the hierarchical DHS data, two-level explanatory variables (individual and community) were used to identify the determinants of early neonatal mortality. The individual-level variables included maternal age, maternal education, place of delivery, birth weight, type of pregnancy, mode of delivery, household wealth index, birth order, type of pregnancy, antenatal care (ANC) utilization, and pregnancy complications. In this study, missing values have been managed by dropping variables assigned as don't know and unspecified numbers.

Place of residence, country, community illiteracy level, and community poverty level make up the community-level variables. The individual-level variables of maternal education and household wealth index, respectively, were aggregated to determine the community levels of illiteracy and poverty.

## Statistical analysis

Once extracted from the source, the data from each country has been appended together to create one single file dataset using Stata 14. To draw valid inferences, the weighting was done using the weighting variables [sample weight (v005), primary sampling unit (v023), and stratum (v021)]. Because DHS data is hierarchical in nature, a multilevel mixed effect logistic regression model was applied to assess determinants of early neonatal mortality in SSA.

Multilevel mixed effect logistic regression has four models; the null model, model I, model II, and model III. Null model is used to determine the applicability of multilevel to the data by determining measure of variation. Model I uses to determine the effect of only individual level factors on early neonatal mortality. Model II uses to determine the effect of community level factors on early neonatal mortality. The last model (model III) shows the effects of both individual and community level factors on the mortality. Multilevel logistic regression is equated as follows,

$$\log\left(\frac{\Pi ij}{1 - \Pi ij}\right) = \beta o + \beta 1x1ij + \cdots + \beta nxnij + uoj + eij$$

Where; πij is the probability of baby to die within one week of life, (1-πij) is the probability of baby not to die within one week of life, βo is log odds of the intercept, β1, . . . βn are effect sizes of individual and community-level factors, x1ij . . . xnij are independent variables of individuals and communities. The quantities uoj and eij are random errors at cluster levels and individual levels respectively.

The measurement of variation (random effect) was done using the median odds ratio (MOR), intra-cluster correlation coefficient (ICC), and proportional change in variation (PCV). The MOR is the unexplained heterogeneity of early neonatal mortality between clusters, and it shows the odds of variation in early neonatal mortality between high- and low-risk clusters, taking two clusters at random. ICC indicates the percent of variation in early neonatal mortality between clusters, and PCV is the percent of variation in early neonatal mortality attributed to individual and community-level factors. The parameters MOR, ICC, and PCV were determined by the following equations: MOR = exp.$(0.95\sqrt{VC})$, $ICC = \frac{VC}{vC} + 3.29 \times 100\%$ and $PCV = \frac{VNull-VC}{VNull} \times 100\%$, where; VC = variance of the cluster for respective model and Null = variance of the null model.

The measure of association or fixed effect was determined by the computation of the AOR at 95% CI and p-value. The best-fit model was selected by using deviance and log likelihood ratio (LLR). A model with a small value of deviance (-2×LLR) and a large value of log likelihood ratio was taken as the best fit.

### Ethical consideration

The data for this study were extracted from the recent DHS dataset. The only prerequisite for access to the DHS data is registration. Therefore, receiving ethical approval for this study was not appropriate. More details regarding DHS data and ethical standards are available online at (http://www.dhsprogram.org.com).

## Results

### Socio-demographic characteristics of the study subjects

A total of 262,763 weighted live births were enrolled in the analysis of this study to determine the death of neonates within the first week of life in sub-Saharan Africa. The mean (±SD) age of mothers of babies who were study subjects was 29±7 years. Nearly three-fourths (75.29%) of babies were born to mothers in the age group of 20–35 years. One hundred twenty-four thousand and seven hundred sixty-two (47.48%) of the study subjects were from households of poor wealth status. About 70% of study subjects resided in rural areas of sub-Saharan Africa. Out of the total live births included in the study, 112,021 (42.63%) were from the eastern part of Africa (Table 2).

### Prevalence of early neonatal mortality in sub-Saharan Africa

Prevalence of early neonatal mortality in sub-Saharan Africa was found to be 22.94 deaths per 1000 live births at 95% CI (22.38, 23.5). Early neonatal mortality in urban and rural areas of sub-Saharan Africa was 22.32 and 23.21 deaths per 1000 live births (Fig 1).

### Measure of variation and model fit statistics

As shown in Table 3 below, 10% of the variation in early neonatal mortality happened between clusters, and the rest of the variation occurred within clusters (ICC = 10%). The MOR in the null model shows that the odds of early neonatal mortality were 1.77 times higher among clusters of high risk for mortality compared to clusters of low risk for mortality. In model I, about 43% of the variation in early neonatal mortality was attributed to individual-level factors (PCV = 42.5%). In the final model, 67% of the variation in early neonatal mortality was due to both individual and community-level factors included in the regression. The odds of early neonatal mortality were 1.39 times higher among high-risk clusters compared to low-risk clusters (MOR = 1.39 in model III). According to the model fit statistics of this study, model III had the lowest deviance and a large LLR value, which indicated that it was the best fitted model.

### Measure of association of early neonatal mortality in sub-Saharan Africa

A total of 14 variables (individual and community) were included in final model (Model III) to identify determinants of early neonatal mortality in SSA. Namely; maternal age, maternal education, birth weight, household wealth index, ANC visits, pregnancy complications, place of delivery, type of pregnancy, birth order, mode of delivery, place of residence, country category, community level illiteracy, community level poverty (Table 4).

The output result of the final model (Model III) of multilevel mixed effect logistic regression shows that maternal age, birth weight, number of ANC visits, mode of delivery, pregnancy complications, multiple pregnancy and community level poverty were significantly associated with early neonatal mortality (Table 4).

The odds of early neonatal mortality were 1.77 times higher for babies born to mothers in the age group of 36–49 (AOR = 1.77, 95% CI: 1.34–2.33) compared to babies born to mothers in the age group of 25–35. For low-birth-weight babies, the odds of early neonatal mortality

**Table 2. Socio-demographic characteristics of the study subjects.**

| Variables | Category | Frequency(n) | Percent (%) |
|---|---|---|---|
| **Individual level variables** | | | |
| Maternal age | 15–19 | 15,598 | 5.94 |
| | 20–35 | 197,846 | 75.29 |
| | 36–49 | 49,319 | 18.77 |
| Maternal education | Unable to read and write | 102,849 | 39.14 |
| | Primary | 89,884 | 34.21 |
| | Secondary and above | 70,028 | 26.65 |
| Birth weight | Low | 13,225 | 9.68 |
| | Normal | 103,531 | 75.79 |
| | High | 19,843 | 14.53 |
| Household wealth index | Poor | 124,762 | 47.48 |
| | Middle | 52,216 | 19.87 |
| | Rich | 85,785 | 32.65 |
| ANC visits | <4 visits | 77,104 | 43.26 |
| | ≥4 visits | 101,134 | 56.74 |
| Birth order | First | 57,299 | 21.81 |
| | Second | 50,346 | 19.16 |
| | Third | 42,245 | 16.08 |
| | Fourth or more | 112,873 | 42.96 |
| Mode of delivery | Cesarean section | 13,085 | 4.99 |
| | Vaginal | 249,125 | 95.01 |
| Pregnancy complications | Yes | 16,805 | 58.72 |
| | No | 11,815 | 41.28 |
| Place of delivery | Home | 94,244 | 36.35 |
| | Health facility | 165,046 | 63.65 |
| Multiple pregnancy | Single | 253,579 | 96.5 |
| | Multiple | 9,184 | 3.5 |
| **Community level variables** | | | |
| Residence | Urban | 79,400 | 30.22 |
| | Rural | 183,363 | 69.78 |
| Country category | Central Africa | 42,678 | 16.24 |
| | East Africa | 112,021 | 42.63 |
| | West Africa | 101,378 | 38.58 |
| | Southern Africa | 6,686 | 2.54 |
| Community level illiteracy | Low | 114,071 | 43.41 |
| | High | 148,692 | 56.59 |
| Community level poverty | Low | 128,978 | 49.09 |
| | High | 133,785 | 50.91 |

were 3.27 times higher (AOR = 3.27, 95% CI: 2.16, 4.94) compared to those born with normal birth weight. The odds of death within the first week of life were 1.12 times higher for babies born to mothers who had less than four ANC visits during pregnancy (AOR = 1.12, 95% CI: 1.01, 1.33), taking babies born to mothers who had four and above ANC visits as a reference.

For babies delivered by caesarean section, the odds of early neonatal mortality were 1.81 (AOR = 1.81; 95% CI: 1.30–2.5) times higher compared to babies delivered vaginally. In reference to babies born to mothers with complicated pregnancies, the odds of early neonatal mortality among babies born with no complications during pregnancy were reduced by 24%

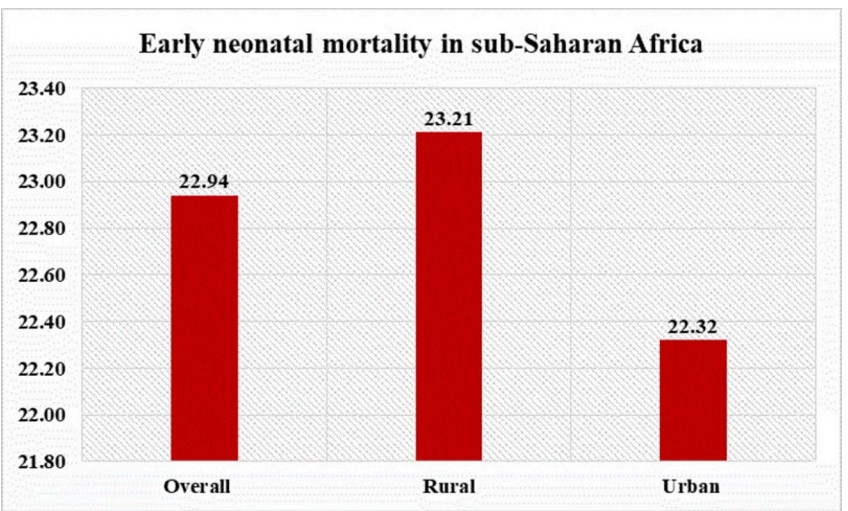

**Fig 1. Prevalence of early neonatal mortality in sub-Saharan Africa.**

(AOR = 0.76, 95% CI: 0.61, 0.94). The odds of early neonatal mortality were 4.02 times higher (AOR = 4.02, 95% CI: 2.66, 6.08) for neonates born multiple compared to those born singleton. Taking babies born in communities with low levels of poverty as reference, the odds of early neonatal mortality were 1.32 (AOR = 1.32, 95% CI: 1.05–1.65) times higher among babies born in communities with high levels of poverty (Table 4).

## Discussion

The death of neonates in the early stages of their lives is a global public health concern, especially in developing countries like sub-Saharan Africa. Using data from the recent demographic and health survey, this study determined the prevalence and associated factors of early neonatal mortality in sub-Saharan Africa.

The prevalence of early neonatal mortality in this study was found to be 22.94 deaths per 1000 live births at a 95% CI (22.38, 23.5). This implies that the rate of early neonatal mortality in sub-Saharan Africa was about twofold higher than the Sustainable Development Goal (SDG) of 2030 to reduce the rate of neonatal mortality to 12 or less per 1000 live births [8].

On the one hand, the prevalence of early neonatal mortality in our study was lower than the prevalence in Nigeria, with 32 deaths per 1000 live births. This lower prevalence of early

**Table 3. Measure of variation and model fit statistics of early neonatal mortality in sub-Saharan Africa.**

| Parameter | Null model | Model I | Model II. | Model III |
|---|---|---|---|---|
| **Measure of variation** | | | | |
| **Variance** | 0.362162 | 0.2079343 | 0.254229 | 0.1199074 |
| **ICC** | 10.0% | 6.0% | 7% | 4% |
| **MOR** | 1.77 | 1.54 | 1.61 | 1.39 |
| **PCV** | Reference | 42.5% | 29.8% | 67.1% |
| **Model fitness** | | | | |
| **LLR** | -28701.5 | -1892.4 | -28629.9 | -1877.4 |
| **Deviance** | 57403.0 | 3784.8 | 57259.8 | 3754.8 |

ICC: intra-cluster correlation coefficient, MO: median odds ratio, PCV: proportional change in variance and LLR: log likelihood ratio.

**Table 4. Measure of association of early neonatal mortality in sub-Saharan Africa.**

| Individual and community level factors | | Model I AOR(95% CI) | Model II AOR(95% CI) | Model III AOR(95% CI) |
|---|---|---|---|---|
| Maternal age | 15–19 | 1.12 (0.88, 1.44) | | 1.30(0.20, 1.94) |
| | 20–35 | 1 | | 1 |
| | 36–49 | 1.53 (1.30, 1.80) | | 1.77(1.34, 2.33)* |
| Maternal education | No formal education | 0.80 (0.67, 0.95) | | 0.83 (0.69, 1.01) |
| | Primary | 0.88 (0.74, 1.04) | | 0.92 (0.77, 1.09) |
| | Secondry and above | 1 | | 1 |
| Birth weight | Low | 3.13 (2.53, 3.90) | | 3.27 (2.16, 4.94)* |
| | Normal | 1 | | 1 |
| | High | 2.37 (0.85, 3.02) | | 2.96 (0.94, 4.51) |
| Household wealth index | Poor | 0.91 (0.78, 1.05)) | | 1.12 (0.84, 1.51) |
| | Middle | 1.05 (0.88, 1.25) | | 1.16 (0.85, 1.6) |
| | Rich | 1 | | 1 |
| ANC visits | <4 visits | 1.15 (1.01, 1.31)) | | 1.14 (0.91, 1.42)* |
| | ≥4 visits | 1 | | 1 |
| Pregnancy complications | Yes | 1 | | 1 |
| | No | 1.01 (0.90, 1.15) | | 0.76 (0.61, 0.94)* |
| Place of delivery | Home | 3.8 (0.83, 4.02) | | 3.59 (0.73, 4.73) |
| | Facility | 1 | | 1 |
| Multiple Pregnancy | Multiple | 4.65 (3.74, 5.83) | | 4.02 (2.66, 6.08)* |
| | Single | 1 | | 1 |
| Birth order | First | 1.29 (0.92, 1.81) | | 1.25 (0.89, 1.769) |
| | Second | 1.01 (0.72, 1.42) | | 0.99 (0.70, 1.39) |
| | Third | 1.01 (0.71, 1.41) | | 0.99 (0.71, 1.39) |
| | Fourth or more | 1 | | 1 |
| Mode of delivery | Cesarean section | 2.05 (1.68, 2.51) | | 1.81 (1.30, 2.50)* |
| | Vaginal | 1 | | 1 |
| Place of residence | Urban | | 1 | 1 |
| | Rural | | 1.05 (1.01, 1.11) | 1.11 (0.85, 1.46) |
| Part of sub-Saharan Africa | Central Africa | | 1.08 (1.01, 1.15) | 1.17 (0.88, 1.57) |
| | East Africa | | 1 | 1 |
| | West Africa | | 1.10 (0.96, 1.26) | 1.09 (0.81, 1.45) |
| | Southern Africa | | 1.24 (0.7, 1.41) | 2.60 (0.8, 3.76) |
| Community level illitracy | Low | | 1 | 1 |
| | High | | 1.04 (0.98, 1.11) | 0.90(0.71, 1.13) |
| Community level poverty | Low | | 1 | 1 |
| | High | | 1.01 (0.94, 1.065) | 1.32 (1.05, 1.65)* |

ANC: antenatal care, CS: caesarean section

*: level of significance (p value) less than 0.05.

neonatal mortality than in the Nigerian study could be related to the DHS data used. Our study was conducted using DHS data from 2014 to 2020, whereas the study in Nigeria used DHS 2013. In this developing world, the rate of mortality among neonates decreases with time due to the advancement of technology and quality maternal and child health services. The more recent the DHS data is, the lower the rate of mortality could be.

On the other hand, the prevalence in this study was higher than the study in Afghanistan with prevalence of 14 deaths per 1000 live births [2]. This discrepancy between prevalence in

our study and Afghanistan could be due to the variation in health infrastructure and socio-economic status across countries. In addition, according to 2018 United Nations estimates, Afghanistan has witnessed about a 50% and 62% reduction in maternal and child mortality from 1990 to 2017, respectively.

The multivariable multilevel mixed effect logistic regression analysis of this study revealed that maternal age, birth weight, ANC visit, multiple pregnancies, mode of delivery, pregnancy complications, and community level of poverty were significantly associated with early neonatal mortality.

The odds of early neonatal mortality were higher for babies born to mothers aged over 35 years compared to babies born to mothers in age group of 25–35 years. Previous studies have also reported the same [2, 20]. The significant association between early neonatal mortality and maternal age above 35 could be due to the fact that low-birth-weight and fetal macrosomia is more prevalent among baies born to mothers of advanced ages [20, 21].

Regarding birth weight, the odds of early neonatal death were higher among low-birth-weight babies compared to babies of normal birth weight. This is consistent with previous studies [21, 22]. One of the common reasons for higher odds of early neonatal mortality among low birth weight babies is that most of the time, low birth weight babies are preterm births and/or small for gestational age [21]. These findings suggest the need to improve mother care during pregnancy, childbirth, and postnatal periods, particularly for low birth weight babies. The World Health Organisation has also developed clinical guidelines to increase baby survival and advocates the need for carefull essential newborn care for low-birth-weight babies [23]. Kangaroo mother care can be an option for neonatal care with low birth weights; it involves skin-to-skin contact between a mother and her newborn [24].

The antenatal care visits during pregnancy of this birth was significantly associated with early neonatal mortality. The odds of death within the first week of life were higher for babies born to mothers who had less than four ANC visits during pregnancy, taking babies born to mothers who had four and above ANC visits as a reference group. It was evidenced that ANC service utilization only reduces mortality of neonates by an estimate of 10–20% [25, 26], although there is insufficient utilization of ANC services in sub-Saharan Africa [27]. A study conducted to determine the impact of ANC on neonatal mortality also found that "the utilisation of at least one antenatal care visit by a skilled provider during pregnancy reduces the risk of neonatal mortality by 39% in sub-Saharan African countries' [28].

Surprisingly, this study found that cesarean section as a mode of delivery was significantly associated with early neonatal mortality. Though a cesarean section is performed to save the life of the newborn, the odds of early neonatal mortality were higher among babies delivered by cesarean section compared to babies delivered vaginally. This finding was coherent with the previous studies [29–31]. On this basis, previous scientific literature also recommends avoiding cesarean sections as intrapartum interventions when there is no clear medical indication that they will improve the outcome for the mother or the baby [30, 32, 33]. However, it could be plausible that the higher odds of early neonatal mortality among babies born via cesarean section were due to the fact that the majority of CS deliveries occur as a last option for delivery when there are pregnancy complications. The association between pregnancy complications and early neonatal mortality in our study supports this justification.

In this study, pregnancy comlication was another factor significantly associated with early neonatal mortality. In reference to babies born to mothers with complicated pregnancies, the odds of early neonatal mortality among babies born with no complications during pregnancy were reduced by 24%. This finding was supported by previous studies [34, 35]. Hence, prevention and carefully tackling complications that happen during pregnancy could be essential strategies to reduce mortality in newborns.

Furthermore, multiple pregnancy was significantly associated with early neonatal mortality. The odds of early neonatal mortality was four times higher for neonates born multiple compared to those born singleton. This finding agrees with previous study [36]. It could be logical explanation that babies born from multiple pregnancies typically suffer growth constraints, low Apgar scores, and very low birth weights [37, 38]. Moreover, multiple pregnancies are more likely to be complicated during pregnancy, labor and after delivery [38].

Eventually, community poverty level was significantly associated with early neonatal mortality. Accordingly, the odds of early neonatal mortality increased by 32 percent among babies born in communities with high levels of poverty, taking babies born in communities with low levels of poverty as a reference group. This finding was supported by previous evidence [39–41]. Babies who die in early neonatal life suffer from conditions and diseases associated with a lack of quality care at or immediately after birth and in the first week of life [42]. Poverty can exacerbate these issues by limiting access to quality healthcare, nutrition, and sanitation, which can increase the risk of neonatal mortality [42].

The use of nationally collected large samples of recent demographic and health survey data from sub-Saharan African countries and the application of an appropriate advanced multilevel mixed effect model were the strengths of this study. However, a few sub-Saharan African countries that had not had a demographic or health survey since 2014 were not included, which may have affected the generalizability of our finding. In addition, the study used DHS data collected in different years, which might have exposed the findings of the study to the differential time effect.

## Conclusion

This study found that about twenty-three neonates out of one thousand live births died within the first week of life in sub-Saharan Africa. The age of mothers, birth weight, antenatal care service utilization, mode of delivery, multiple pregnancy, complications during pregnancy, and community poverty should be considered while designing policies and strategies targeting early neonatal mortality in sub-Saharan Africa.

## Author Contributions

**Conceptualization:** Yirgalem Mohammed, Alemneh Tadesse Kassie, Alebachew Ferede Zegeye.

**Formal analysis:** Tadesse Tarik Tamir.

**Methodology:** Tadesse Tarik Tamir, Yirgalem Mohammed.

**Software:** Tadesse Tarik Tamir.

**Validation:** Yirgalem Mohammed, Alemneh Tadesse Kassie, Alebachew Ferede Zegeye.

**Visualization:** Alemneh Tadesse Kassie, Alebachew Ferede Zegeye.

**Writing – original draft:** Tadesse Tarik Tamir.

**Writing – review & editing:** Yirgalem Mohammed, Alemneh Tadesse Kassie, Alebachew Ferede Zegeye.

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
