## [Decision Letter · Decision Letter 0]

11 Dec 2023

PONE-D-23-16077Early Neonatal Mortality and Determinants in Sub-Saharan Africa: Multilevel Analysis of Recent Demographic and Health SurveyPLOS ONE

Dear Dr. Tamir,

Thank you for submitting your manuscript to PLOS ONE. After careful consideration, we feel that it has merit but does not fully meet PLOS ONE’s publication criteria as it currently stands. Therefore, we invite you to submit a revised version of the manuscript that addresses the points raised during the review process.

We look forward to receiving your revised manuscript.

Kind regards,

Abera Mersha, MSc.

Academic Editor

PLOS ONE

Journal Requirements:

- [citations]

In your revision ensure you cite all your sources (including your own works), and quote or rephrase any duplicated text outside the methods section. Further consideration is dependent on these concerns being addressed.

Reviewers' comments:

Reviewer's Responses to Questions

**Comments to the Author**

1. Is the manuscript technically sound, and do the data support the conclusions?

Reviewer #1: Partly

Reviewer #2: Yes

2. Has the statistical analysis been performed appropriately and rigorously? 

Reviewer #1: Yes

Reviewer #2: Yes

3. Have the authors made all data underlying the findings in their manuscript fully available?

Reviewer #1: No

Reviewer #2: Yes

4. Is the manuscript presented in an intelligible fashion and written in standard English?

Reviewer #1: No

Reviewer #2: No

5. Review Comments to the Author

Reviewer #1: The work is technically sound though the following need to be addressed;

Methods

• Use of Kids Records is not the most appropriate word to use unless it was officially called that as in statement below;

This study used the Kids Records (KR) dataset to determine the outcome variable.

The two statements are contradictory, address the discrepancy;

• A sample of 262,763 live births were included in the study (Table 1).

Table 1: Sample size for early neonatal mortality in Sub-Saharan Africa countries (n=352,605).

• Be consistent use either subject or participants.

3. Is the methodology feasible and described in sufficient detail to allow the work to be replicable?

There is need to provide more detailed information about the inclusion criteria

• Describe the inclusion criteria in this study and how you deal with missing data which a common problem with secondary data

• Paragraph your work well

• Work on spacing between the reference and the words before

• Ensure that the font size is the same throught the document see the paragraph below

In this study, pregnancy complication was another factor significantly associated with early neonatal mortality. In reference to babies born to mothers with complicated pregnancies, the odds of early neonatal mortality among babies born with no complications during pregnancy were reduced by 24%. This finding was supported by previous studies (29, 30)

Introduction

• Number the lines so that they are easy to refer to

• Be specific for example when you stated in the introduction that;

• Before turning five, 3.3 million more children will die(4). Specify what measurement five is … days, months..years etc

• Please talk of SDG because these are the ones currently being referred to ..

• There many grammatically errors

• Provide references for the paragraph below;

• The early neonatal period, the first seven days of life, are the most precarious for a baby's survival. Children face the highest risk of death in their first week of life, especially in low- and middle income countries.

• Provide references to qualify the paragraph below;

Since neonatal mortality in the first week of life is understudied in sub-Saharan

Africa, coming up with its magnitude and determinants helps understand the current status of

Results

• Clarify on what participants ( mothers or neonates) with mean/SD of 29±7 standard deviation (SD). you were referring to in the statement below

neonates within the first week of life in sub-Saharan Africa. The mean age of participants was

29±7 standard deviation (SD). Nearly three-fourths (75.29%) of babies in this study were born to mothers in the age group of 20–35 years. One hundred twenty-four thousand and seven hundred

• State the units of measurement used in The mean age of participants was 29±7 standard deviation (SD).

• Figure 1 is missing

• Bold the titles of the tables so that they can easily be identified….

• Leave space between the narratives before the tables and the tables

• Spelling of maternal education in table 4

• Table 4 should reflect the outcome as in two by two format. Ie outcome vs independent variables under the different models

Reviewer #2: The authors have chosen an important public health problem for their study. The evidence from the study will add to the knowledge base which will have contribution to policy makers and program implementers working to reduce neonatal mortality. Here below, please find my review comments:

Abstract

• Please report the specific category within a variable that is a risk factor than mentioning the entire variable as a risk factor E.g. ANC visits, mode of delivery etc.

Introduction:

• ‘’The neonatal period (the first 28 days of life) accounted for more than 44 percent of the estimated 6.3 million child fatalities worldwide in 2032, while the early neonatal period accounted for 75% of these neonatal deaths’’. Please make sure you use the right tense in the above sentence. A 2032 newborn mortality is a prediction. It is better if you can describe the current burden of neonatal mortality and its contribution.

• It is better to bring up the definition of early neonatal mortality upfront to understand your paper well

• ‘’In Africa, 1.16 million babies perish within the first month of life; half of them die on their very first day. Before turning five, 3.3 million more children will die (4)’’. From this statement, one can calculate the contribution of neonatal mortality in Africa which is roughly 35% of the under-five deaths in the continent. This is far below the 44% contribution for the 2032 worldwide prediction. The truth is, despite still unacceptable magnitude, neonatal mortality is on a declining trend worldwide. This also again contradicts with the contribution of neonatal mortality to under five mortalities mentioned in the follow on paragraph (40.3). Please double check the credibility of your sources and the accuracy of your references.

• In the following sentences the authors talk a lot about MDG while it is a long gone target replaced by the sustainable development goal (SDG). Better to mention the targets and progress of the SDG about neonatal mortality.

Methods

• Please put the countries in Table 1 in alphabetical order (E.g. Chad)

• Aggregating individual level variables to use them as group level variables will cause atomistic fallacy. One of the advantage of using mixed effects modeling is to avoid both ecological and atomistic fallacies. Using group level variables as individual variables and vice versa will cause fallacies. Hence, I suggest you treat maternal education and wealth index at their proper level.

• In the calculation of the prevalence of early neonatal mortality, you need to use weighting. It needs to be also described in the analysis section.

Result

• When you report measures of central tendency please report them as for example mean (+SD) followed by the actual numbers.

• Mode of Delivery in Table 2 is stated as ‘Yes’’ and ‘’No’’ compared to Table 4 (CS Vs vaginal). What do ‘’Yes’’ and ‘’No’’ categories in Table 2 stand for?

Discussion

• In the discussion part, it is good to compare apples with apples. The mortality prevalence is a result for SSA and the comparison is better made with similar sub-continent levels than a single country.

• You reported ‘’ The number of antenatal care visits during pregnancy of this birth was significantly associated with early neonatal mortality’’. However, your data was categorical. I think you need to correct the statement.

• The risk of neonatal mortality is high among babies born using CS. What is the implication of more deaths in CS than vaginal deliveries? Can it be related to the quality of CS service? How can CS meant to save the lives of the mother and the child cause more deaths? Could this be related to the fact that already troubled pregnancy undergo CS? Can you compare this result with other studies? You have described it somehow, but needs more elaboration.

General comment

• Please spell out acronyms the first time they appear in the body text. (E.g. MDG, MOR, ICC, PCV, R etc). Plus there are a lot of language errors and words that are misspelt. Please review accordingly.

• You need to edit the language and the flow of ideas very well.

6. PLOS authors have the option to publish the peer review history of their article (what does this mean?). If published, this will include your full peer review and any attached files.

Reviewer #1: No

Reviewer #2: No

---

## [Author Response · Author response to Decision Letter 0]

14 Dec 2023

Response to comments

Manuscript ID: PONE-D-23-16077

Title: Early neonatal mortality and determinants in sub-Saharan Africa: findings from recent Demographic and Health Survey data

Journal: PLOS ONE

Subject: Submission of revised manuscript

We thank the editors facilitating this manuscript for possible publication and the reviewers for their time spent reviewing the manuscript and their insightful comments and suggestions that helped improve the quality of this manuscript. The comments are encouraging, and the reviewer appears to share our judgment that this study and its findings are significant from a scientific standpoint. Please see our thorough answer to the comments below. Please refer to the revised manuscript file that has been supplied separately in the attachment.

Yours sincerely,

Tadesse Tarik Tamir, corresponding author (on behalf of all authors)

University of Gondar, Gondar, Ethiopia

Response to reviewers’ comments 

Reviewer #1: The work is technically sound though the following need to be addressed;

Response: Dear reviewer, first and foremost, we appreciate your enthusiasm for our manuscript's subject and hypotheses, as well as your detailed perspectives and insightful comments.

Methods

• Use of Kids Records is not the most appropriate word to use unless it was officially called that as in statement below;

This study used the Kids Records (KR) dataset to determine the outcome variable.

Response: Dear reviewer, Thank you for the comments and suggestions. In the DHS program, the standard name for children's data is Kids Record (KR). We mean to provide information to readers that helps them understand which record the authors used to determine the aim of the study.

The two statements are contradictory, address the discrepancy;

• A sample of 262,763 live births were included in the study (Table 1).

Table 1: Sample size for early neonatal mortality in Sub-Saharan Africa countries (n=352,605).

Response: Dear reviewer, sorry for inconvenience. The discrepancy was just due to typing error. We addressed the point accordingly. Kindly find the point in our revised manuscript. 

• Be consistent use either subject or participants.

Response: Dear reviewer, thank you for the insightful comments and suggestions. We affirm that the comments are fully addressed as per your direction. Kindly find the point in our updated manuscript. 

3. Is the methodology feasible and described in sufficient detail to allow the work to be replicable?

There is need to provide more detailed information about the inclusion criteria

• Describe the inclusion criteria in this study and how you deal with missing data which a common problem with secondary data

Response: Dear reviewer, Thank you for your concerns about the inclusion criteria and the missing data. The study and source population of this study are provided in the revised manuscript. Kindly see the revised manuscript. Regarding the inclusion criteria, this study used secondary data, and it was not the authors who included and excluded the respondents and study subjects.

• Paragraph your work well

Response: Comments accepted and addressed accordingly. Kindly find the point in our revised manuscript. 

• Work on spacing between the reference and the words before

Response: Dear reviewer, we addressed the comments accordingly. Kindly see point in our revised manuscript. 

• Ensure that the font size is the same throught the document see the paragraph below

In this study, pregnancy complication was another factor significantly associated with early neonatal mortality. In reference to babies born to mothers with complicated pregnancies, the odds of early neonatal mortality among babies born with no complications during pregnancy were reduced by 24%. This finding was supported by previous studies (29, 30).

Response: Dear reviewer, thank you for the comment and suggestions. We addressed the comments accordingly. Please see the point in our revised manuscript. 

Introduction

• Number the lines so that they are easy to refer to

Response: Dear reviewer, the revised manuscript is provided with the line numbers. Thank you. 

• Be specific for example when you stated in the introduction that;

• Before turning five, 3.3 million more children will die(4). Specify what measurement five is … days, months..years etc

Response: Dear reviewer, thank you for the feedback. We mean five years. The point is now addressed in our revised work. Kindly see the point in our revised manuscript. 

• Please talk of SDG because these are the ones currently being referred to ..

Response: Dear reviewer, thank you for such a scientifically sound comments. We incorporated your comment into our revised manuscript. Kindly find the point in the revised manuscript.

• There many grammatically errors

Response: Dear reviewer, taking your feedback into account, we thoroughly made modification to our write up. Kindly see our modifications in the revised manuscript. 

• Provide references for the paragraph below;

• The early neonatal period, the first seven days of life, are the most precarious for a baby's survival. Children face the highest risk of death in their first week of life, especially in low- and middle income countries.

Response: Dear reviewer, thank you for your invaluable comments. We provided our revised manuscript with reference to the particular information. Kindly find the point in our revised manuscript. 

• Provide references to qualify the paragraph below;

Since neonatal mortality in the first week of life is understudied in sub-Saharan

Africa, coming up with its magnitude and determinants helps understand the current status of

Response: Dear reviewer, thank you for your suggestions. We made modifications to the paragraph in revised manuscript. Kindly see it in our revised manuscript. 

Results

• Clarify on what participants (mothers or neonates) with mean/SD of 29±7 standard deviation (SD). You were referring to in the statement below 

neonates within the first week of life in sub-Saharan Africa. The mean age of participants was 29±7 standard deviation (SD). Nearly three-fourths (75.29%) of babies in this study were born to mothers in the age group of 20–35 years. One hundred twenty-four thousand and seven hundred

Response: Dear reviewer, Thank you for the feedback on the section, which needs to clearly state what the authors mean to transmit to the readers. We mean the mean (±SD) of the age of mothers of neonates. The point is now addressed accordingly. Kindly find the revised version of the manuscript.

• State the units of measurement used in The mean age of participants was 29±7 standard deviation (SD).

Response: Dear reviewer, thank you for such a scientifically guiding comments on our results. We incorporated your comments in to our revised manuscript. Kindly find the point in our revised manuscript.

• Figure 1 is missing

Response: Dear reviewer. Figure 1 was separately uploaded to the journal. Kindly find the separately uploaded figure in our revised submission.

• Bold the titles of the tables so that they can easily be identified….

Response: Dear reviewer, thank you for the comments. We incorporated your comments in to our revised manuscript. Kindly find the point in our revised manuscript. 

• Leave space between the narratives before the tables and the tables

Response: Dear reviewer, thank you for the guidance. We addressed the point as per your direction. Kindly see the revised manuscript.

• Spelling of maternal education in table 4

Response: Dear reviewer, appreciating your in-depth review of our manuscript, we corrected the spelling of the variable. Please find the point in our revised manuscript. 

• Table 4 should reflect the outcome as in two by two format. Ie outcome vs independent variables under the different models

Reviewer: Dear reviewer, thank you for your insightful comments and suggestions. Due to the multilevel nature of the mixed effect models and should be compared together, we couldn’t incorporate the crude outcome status of each independent variables. 

Once again, we thank you for your insightful comments, suggestions, and guidance, which have significantly improved the study.

Reviewer #2: The authors have chosen an important public health problem for their study. The evidence from the study will add to the knowledge base which will have contribution to policy makers and program implementers working to reduce neonatal mortality. Here below, please find my review comments:

Response: Dear reviewer, first of all, we appreciate your enthusiasm for our manuscript's subject and hypotheses, as well as your detailed perspectives and insightful comments.

Abstract

• Please report the specific category within a variable that is a risk factor than mentioning the entire variable as a risk factor E.g. ANC visits, mode of delivery etc.

Response: Dear reviewer, Thank you for your insightful comments. We addressed the comment accordingly. Kindly find the point in the abstract section of our revised manuscript. 

Introduction:

• ‘’The neonatal period (the first 28 days of life) accounted for more than 44 percent of the estimated 6.3 million child fatalities worldwide in 2032, while the early neonatal period accounted for 75% of these neonatal deaths’’. Please make sure you use the right tense in the above sentence. A 2032 newborn mortality is a prediction. It is better if you can describe the current burden of neonatal mortality and its contribution.

• It is better to bring up the definition of early neonatal mortality upfront to understand your paper well

Response: Dear reviewer, thank you for your insightful suggestion. We provided the revised manuscript with the definition of early neonatal mortality upfront in the introduction as per your comment. Kindly find the point in the abstract section of our revised manuscript.

 • ‘’In Africa, 1.16 million babies perish within the first month of life; half of them die on their very first day. Before turning five, 3.3 million more children will die (4)’’. From this statement, one can calculate the contribution of neonatal mortality in Africa which is roughly 35% of the under-five deaths in the continent. This is far below the 44% contribution for the 2032 worldwide prediction. The truth is, despite still unacceptable magnitude, neonatal mortality is on a declining trend worldwide. This also again contradicts with the contribution of neonatal mortality to under five mortalities mentioned in the follow on paragraph (40.3). Please double check the credibility of your sources and the accuracy of your references.

Response: Dear reviewer, taking your comments into account, we made modifications and adjustments to the introduction of our manuscript. Kindly find the point in our revised manuscript. Thank you.

• In the following sentences the authors talk a lot about MDG while it is a long gone target replaced by the sustainable development goal (SDG). Better to mention the targets and progress of the SDG about neonatal mortality.

Response: Dear reviewer, thank you for your constructive comment. The comment is now addressed. Kindly find the point in our updated document. 

Methods

• Please put the countries in Table 1 in alphabetical order (E.g. Chad)

Response: Dear reviewer, thank you for your constructive feedback. The countries are now ordered alphabetically. 

• Aggregating individual level variables to use them as group level variables will cause atomistic fallacy. One of the advantage of using mixed effects modeling is to avoid both ecological and atomistic fallacies. Using group level variables as individual variables and vice versa will cause fallacies. Hence, I suggest you treat maternal education and wealth index at their proper level.

Response: Dear reviewer, thank you for such a constructive view of expertise. We just aggregated wealth index and maternal education by considering that the mixed effect model could control for the fallacies. Originally, the authors used this technique by looking previous literatures. Taking your comments into account, now on the authors will treat individual and group variables at their proper level. Thank you.

• In the calculation of the prevalence of early neonatal mortality, you need to use weighting. It needs to be also described in the analysis section.

Response: Dear reviewer, Thank you for the concern. The computation of an outcome in DHS should always be supported by appropriate weighting. Though the prevalence of early neonatal mortality reported in our manuscript was weighted, we missed describing it in the method section. Taking your comment into consideration, we have now mentioned the point in our updated work. Kindly see the point in our revised manuscript.

Result

• When you report measures of central tendency please report them as for example mean (+SD) followed by the actual numbers.

Response: Dear reviewer, thank you for your guiding comments. We addressed it as per your direction. 

• Mode of Delivery in Table 2 is stated as ‘Yes’’ and ‘’No’’ compared to Table 4 (CS Vs vaginal). What do ‘’Yes’’ and ‘’No’’ categories in Table 2 stand for?

Response: Dear reviewer, Thank you for your invaluable scientific guidance. ’Yes’’ and ‘’No’’ categories in Table 2 stand for CS and not CS (Vaginal) modes of delivery, respectively. Sorry for the discrepancy. We addressed the discrepancy in our revised manuscript. 

Discussion

• In the discussion part, it is good to compare apples with apples. The mortality prevalence is a result for SSA and the comparison is better made with similar sub-continent levels than a single country.

Response: Dear reviewer, Thank you for your insightful feedback on our discussion. You are correct; general studies should be discussed with similar kinds of studies instead of pocket ones. We made some amendments to the section as per your comment, using information available to the best of our search. We discussed our prevalence with prevalence from similar kinds of data (DHS), though some are not intercontinental. We faced the challenge of a lack of studies with similar aims at similar levels. Thank you!

• You reported ‘’ The number of antenatal care visits during pregnancy of this birth was significantly associated with early neonatal mortality’’. However, your data was categorical. I think you need to correct the statement.

Response: Thank you, the concern is addressed accordingly. 

• The risk of neonatal mortality is high among babies born using CS. What is the implication of more deaths in CS than vaginal deliveries? Can it be related to the quality of CS service? How can CS meant to save the lives of the mother and the child cause more deaths? Could this be related to the fact that already troubled pregnancy undergo CS? Can you compare this result with other studies? You have described it somehow, but needs more elaboration.

Response: Dear reviewer, thank you for such a scientifically valuable comment. We made modifications to the section by including additional justifications. Kindly find the point in our revised manuscript. 

General comment

• Please spell out acronyms the first time they appear in the body text. (E.g. MDG, MOR, ICC, PCV, R etc). Plus there are a lot of language errors and words that are misspelt. Please review accordingly.

Response: Dear reviewer, we spelled out all the acronyms the first time they appear in the text throughout the manuscript and mentioned them in the abbreviation section of your manuscript. Kindly see the point in our revised manuscript. 

• You need to edit the language and the flow of ideas very well.

Response: Dear reviewer, we thoroughly checked the language and made rigorous modifications to the errors as per necessary. 

Once again, we thank you for your insightful comments, suggestions, and guidance, which have significantly improved the study.

---

## [Decision Letter · Decision Letter 1]

16 Jan 2024

PONE-D-23-16077R1Early neonatal mortality and determinants in sub-Saharan Africa: findings from recent Demographic and Health Survey dataPLOS ONE

Dear Dr. Tamir,

Thank you for submitting your manuscript to PLOS ONE. After careful consideration, we feel that it has merit but does not fully meet PLOS ONE’s publication criteria as it currently stands. Therefore, we invite you to submit a revised version of the manuscript that addresses the points raised during the review process.

We look forward to receiving your revised manuscript.

Kind regards,

Abera Mersha, MSc.

Academic Editor

PLOS ONE

Journal Requirements:

Reviewers' comments:

Reviewer's Responses to Questions

**Comments to the Author**

1. If the authors have adequately addressed your comments raised in a previous round of review and you feel that this manuscript is now acceptable for publication, you may indicate that here to bypass the “Comments to the Author” section, enter your conflict of interest statement in the “Confidential to Editor” section, and submit your "Accept" recommendation.

Reviewer #1: (No Response)

Reviewer #2: All comments have been addressed

2. Is the manuscript technically sound, and do the data support the conclusions?

Reviewer #1: Yes

Reviewer #2: Yes

3. Has the statistical analysis been performed appropriately and rigorously? 

Reviewer #1: Yes

Reviewer #2: Yes

4. Have the authors made all data underlying the findings in their manuscript fully available?

Reviewer #1: Yes

Reviewer #2: Yes

5. Is the manuscript presented in an intelligible fashion and written in standard English?

Reviewer #1: No

Reviewer #2: Yes

6. Review Comments to the Author

Reviewer #1: The sentence numbers should be continuous from title to references rather than breaking them down per section because it can create confusion.

Title ; Capitalise every word that is more than three words in the title

Referencing…. cite ALL the reference numbers in square brackets.

Line 15 of Statistical Analysis … there is need to work on the verb

Once extracted from the source, the data for each country has been assembled, m

• Use of Kids Records is not the most appropriate word to use unless it was officially

called that as in the statement .. The word Kids is used when dealing with animals.

RESULTS

Line 28… it should be Results.

Revise of the categories of ANC visits in table 2 in because they seem to mean the same thing.

ANC visits

>4 visits

≥4 visits

In Table 2, Please write the fourth category of Birth order in words to maintain consistency in

Regarding the Mode of delivery variable in Table 2 and Table 4:…. the category (CS) should be written in full.

In the discussion section…. improve on the discussion of the variable under Line 17 to 21

Limitations… since the data from the different countries was collected in different years, couldn’t this have an effect on the results. if so please include it in the limitations

Reviewer #2: The authors have addressed all of the comments. However, I still think that manuscript can benefit from additional language editing.

7. PLOS authors have the option to publish the peer review history of their article (what does this mean?). If published, this will include your full peer review and any attached files.

Reviewer #1: No

Reviewer #2: No

---

## [Author Response · Author response to Decision Letter 1]

17 Jan 2024

Response to comments

Manuscript ID: PONE-D-23-16077

Title: Early Neonatal Mortality and Determinants in sub-Saharan Africa: Findings from Recent Demographic and Health Survey data

Journal: PLOS ONE

Subject: Submission of revised manuscript

We thank the editors facilitating this manuscript for possible publication and the reviewers for their time spent reviewing the manuscript and their insightful comments and suggestions that helped improve the quality of this manuscript. The comments are encouraging, and the reviewer appears to share our judgment that this study and its findings are significant from a scientific standpoint. Please see our thorough answer to the comments below. Please refer to the revised manuscript file that has been supplied separately in the attachment.

Yours sincerely,

Tadesse Tarik Tamir, corresponding author (on behalf of all authors)

University of Gondar, Gondar, Ethiopia

Response to editors and reviewers’ and comments 

Response: Thank you. The authors considered proofreading the manuscript and corrected any typographical or grammatical errors.

Reviewer #1: The sentence numbers should be continuous from title to references rather than breaking them down per section because it can create confusion. 

Response: Dear reviewer, the revised manuscript is provided with the continuous line numbers. Thank you. 

Title ; Capitalise every word that is more than three words in the title

Response: Thank you. We fully addressed the concern. 

Referencing…. cite ALL the reference numbers in square brackets.

Response: Dear reviewer, taking your comments into account we cited the reference numbers in square brackets. Kindly find the point in our revised manuscript. 

Line 15 of Statistical Analysis … there is need to work on the verb

Once extracted from the source, the data for each country has been assembled, m

Response: Dear reviewer, thank you for the constructive comments. The sentence is now amended for clarity to “Once extracted from the source, the data from each country has been appended together to create one single file dataset using Stata 14.” 

• Use of Kids Records is not the most appropriate word to use unless it was officially

called that as in the statement .. The word Kids is used when dealing with animals.

Response: Dear reviewer, thank you for your insightful comment. We made modifications to the statement in revised manuscript. Thanks to your constructive comment, it is now changed to “This study used the children’s record dataset to determine the outcome variable.” 

RESULTS

Line 28… it should be Results.

Response: Thank you for the comments. We addressed it accordingly. Kindly find it in our currently revised manuscript. 

Revise of the categories of ANC visits in table 2 in because they seem to mean the same thing.

ANC visits

>4 visits

≥4 visits

Response: Dear reviewer, thank you for your correction. The point is now addressed accordingly. 

In Table 2, Please write the fourth category of Birth order in words to maintain consistency in

Response: Dear reviewer, the comment is invaluable and it is now addressed accordingly. Please find it in the revised manuscript. 

Regarding the Mode of delivery variable in Table 2 and Table 4:…. the category (CS) should be written in full.

Response: Thank you, the comment is addressed accordingly. Kindly find the point in our revised manuscript. 

In the discussion section…. improve on the discussion of the variable under Line 17 to 21

Response: Dear reviewer, thank you for your scientifically invaluable feedback. We made improvements to the section as per your comment. Kindly find the point in our revised manuscript. 

Limitations… since the data from the different countries was collected in different years, couldn’t this have an effect on the results. if so please include it in the limitations

Response: Dear reviewer, thank you for such an insightful concern. The concern is now addressed by including it in the limitation section of the revised manuscript. Please see the limitation section of our revised manuscript. 

Reviewer #2: The authors have addressed all of the comments. However, I still think that manuscript can benefit from additional language editing.

Response: Thank you. The authors considered proofreading the manuscript and corrected any typographical or grammatical errors.

We thank you for your insightful comments, suggestions, and guidance, which have significantly improved the study.

---

## [Decision Letter · Decision Letter 2]

20 Feb 2024

PONE-D-23-16077R2Early neonatal mortality and determinants in sub-Saharan Africa: findings from recent Demographic and Health Survey dataPLOS ONE

Dear Dr. Tamir,

Thank you for submitting your manuscript to PLOS ONE. After careful consideration, we feel that it has merit but does not fully meet PLOS ONE’s publication criteria as it currently stands. Therefore, we invite you to submit a revised version of the manuscript that addresses the points raised during the review process.

We look forward to receiving your revised manuscript.

Kind regards,

Abera Mersha, MSc.

Academic Editor

PLOS ONE

Journal Requirements:

Reviewers' comments:

Reviewer's Responses to Questions

**Comments to the Author**

1. If the authors have adequately addressed your comments raised in a previous round of review and you feel that this manuscript is now acceptable for publication, you may indicate that here to bypass the “Comments to the Author” section, enter your conflict of interest statement in the “Confidential to Editor” section, and submit your "Accept" recommendation.

Reviewer #1: (No Response)

2. Is the manuscript technically sound, and do the data support the conclusions?

Reviewer #1: Yes

3. Has the statistical analysis been performed appropriately and rigorously? 

Reviewer #1: Yes

4. Have the authors made all data underlying the findings in their manuscript fully available?

Reviewer #1: (No Response)

5. Is the manuscript presented in an intelligible fashion and written in standard English?

Reviewer #1: No

6. Review Comments to the Author

Reviewer #1: Spacing between paragraphs should be worked on especially in the discussion section.

The spacing between in text references and words in sentences line 249_251

References in the Reference list should follow PLos One recommendations with Authors name being first before author initials

Bold the title for the figure 1.

After those minor corrections i recommend that the work be published

7. PLOS authors have the option to publish the peer review history of their article (what does this mean?). If published, this will include your full peer review and any attached files.

Reviewer #1: No

---

## [Author Response · Author response to Decision Letter 2]

21 Feb 2024

Response to reviewer comments 

The spacing between in text references and words in sentences line 249_251

Response: Dear reviewer, we addressed the comment accordingly. 

References in the Reference list should follow PLos One recommendations with Authors name being first before author initials.

Response: Dear reviewer, we provided the revised manuscript with the list of references as per PLOS ONE recommendation. Thank you!

Bold the title for the figure 1.

Response: Dear reviewer, Thank you. We addressed the concern accordingly.

---

## [Decision Letter · Decision Letter 3]

7 May 2024

Early neonatal mortality and determinants in sub-Saharan Africa: findings from recent Demographic and Health Survey data

PONE-D-23-16077R3

Dear Dr. Tamir,

We’re pleased to inform you that your manuscript has been judged scientifically suitable for publication and will be formally accepted for publication once it meets all outstanding technical requirements.

Kind regards,

Abera Mersha, MSc.

Academic Editor

PLOS ONE

Additional Editor Comments (optional):

Reviewers' comments:

Reviewer's Responses to Questions

**Comments to the Author**

1. If the authors have adequately addressed your comments raised in a previous round of review and you feel that this manuscript is now acceptable for publication, you may indicate that here to bypass the “Comments to the Author” section, enter your conflict of interest statement in the “Confidential to Editor” section, and submit your "Accept" recommendation.

Reviewer #1: All comments have been addressed

2. Is the manuscript technically sound, and do the data support the conclusions?

Reviewer #1: Yes

3. Has the statistical analysis been performed appropriately and rigorously? 

Reviewer #1: (No Response)

4. Have the authors made all data underlying the findings in their manuscript fully available?

Reviewer #1: Yes

5. Is the manuscript presented in an intelligible fashion and written in standard English?

Reviewer #1: Yes

6. Review Comments to the Author

Reviewer #1: Almost all comments have been addressed though paragraphing format should follow the recommended format and made uniform

7. PLOS authors have the option to publish the peer review history of their article (what does this mean?). If published, this will include your full peer review and any attached files.

Reviewer #1: No

---

## [Editor Report · Acceptance letter]

29 May 2024

PONE-D-23-16077R3 

PLOS ONE

Dear Dr. Tamir, 

I'm pleased to inform you that your manuscript has been deemed suitable for publication in PLOS ONE. Congratulations! Your manuscript is now being handed over to our production team.

Kind regards, 

on behalf of

Mr. Abera Mersha 

Academic Editor

PLOS ONE